# Deconfounding Imitation Learning with Variational Inference

**Risto Vuorio**[*†‡ 1,2], **Pim de Haan**[* 2,3], **Johann Brehmer**[2], **Hanno Ackermann**[2], **Daniel Dijkman**[2], **and Taco Cohen**[2]

[1]University of Oxford
[2]Qualcomm AI Research. Qualcomm AI Research is an initiative of Qualcomm Technologies, Inc.
[3]QUVA Lab, University of Amsterdam

**Reviewed on OpenReview:** `https://openreview.net/forum?id=3FsVtsISHW`

## Abstract

Standard imitation learning can fail when the expert demonstrators have different sensory inputs than the imitating agent. This is because partial observability gives rise to hidden confounders in the causal graph. Previously, to work around the confounding problem, policies have been trained by accessing the expert's policy or using inverse reinforcement learning (IRL). However, both approaches have drawbacks as the expert's policy may not be available and IRL can be unstable in practice. Instead, we propose to train a variational inference model to infer the expert's latent information and use it to train a latent-conditional policy. We prove that using this method, under strong assumptions, the identification of the correct imitation learning policy is theoretically possible from expert demonstrations alone. In practice, we focus on a setting with less strong assumptions where we use exploration data for learning the inference model. We show in theory and practice that this algorithm converges to the correct interventional policy, solves the confounding issue, and can under certain assumptions achieve an asymptotically optimal imitation performance.

## 1 Introduction

Successful training of policies via behavioral cloning (BC) requires a high quality expert dataset. The conditions under which the dataset is collected have to exactly match those encountered by the imitator. Sometimes such data collection may not be feasible. For example, imagine collecting data from human drivers for training a driving policy for a self-driving car. The drivers are aware of the weather forecast and lower their speed in icy conditions, even when the ice is not visible on the road. However, if the driving policy in the self-driving car does not have access to the same forecast, it is unaware of the ice on the road and may thus be unable to adapt to the conditions. In this paper, we focus on such imitation learning in settings where the expert knows more about the world than the imitator.

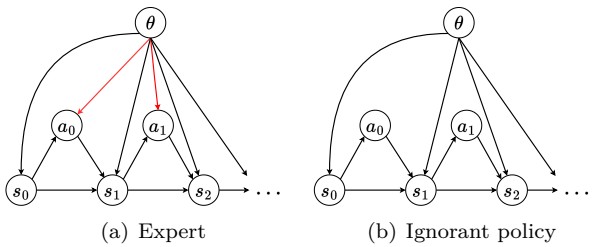

(a) Expert      (b) Ignorant policy

Figure 1: Bayes nets for **(a)** an expert trajectory and **(b)** an imitator trajectory. The expert action depends on the latent variable $\theta$ (red arrows) whereas the imitator action does not.

---

[*]Equal contribution.
[†]Corresponding author: `risto.vuorio@gmail.com`
[‡]Work done during internship at Qualcomm AI Research.

The information the expert observes about the world but the imitator does not can be modeled as a latent variable that affects the dynamics of the environment. Imitating an expert who observes the latent variable, with a policy that does not, results in the learned policy marginalizing its uncertainty about the latent variable. This can result in poor performance, as the agent always acts randomly in states where the expert acts on its knowledge about the latent. To fix this, we assume that the imitator policy is dependent on the entire history of interaction with the environment instead of just the current observation. This enables the agent to make an inference about the value of the latent variable based on everything it has observed so far, which eventually allows it to break ties over the actions in states where the latent variable matters. However, this introduces another problem where the latent variable of the environment acts as a causal confounder in the graph modeling the interaction of the agent with its environment as illustrated in Figure 1.

In such a situation, an imitating agent may take their own past actions as evidence for the values of the confounder. The self-driving car, for instance, could conclude that as it is driving fast there can be no ice on the road. This issue of *causal delusion* was first pointed out in Ortega & Braun (2010a;b) and studied in more depth by Ortega et al. (2021). This problem can be a source of "delusions" in generative models including large language models Ortega et al. (2021). Instead of general generative modeling, we focus on a control setting where we have access to the true environment where the agent is going to be deployed.

Ortega et al. (2021) show that using the classic dataset aggregation (DAgger) algorithm (Ross et al., 2011) solves this problem by querying the expert policy in the new situations the agent encounters to provide new supervision for learning the correct behavior. However, this kind of query access to the expert may not be available in practice. As another solution to the confounded imitation learning problem, Swamy et al. (2022b) propose to use inverse reinforcement learning (IRL) (Russell, 1998; Ng et al., 2000). However, IRL typically requires adversarial methods (Ho & Ermon, 2016; Fu et al., 2017), which are not as well behaved and scalable as supervised learning.

To get around the confounding problem without query access to the expert or IRL, we propose a practical algorithm based on variational inference. In our approach, an inference model for the latent variable is learned from exploration data collected using the imitator's policy. This inference model is used for inferring the latent variable on the expert trajectories for training the imitator and for inferring the latent variable online when the imitator is deployed in the environment. We show that, in theory, this algorithm converges to the expert's policy. In other words, we show that the expert's policy can be identified. Furthermore we show that under strong assumptions this identification can be carried out purely from offline data. Finally, we validate its performance empirically in a set of confounded imitation learning problems with high-dimensional observation and action spaces. These contributions can be summarized as follows:

- We propose a practical method based on variational inference to address confounded IL without expert queries. We verify its performance empirically in various confounded control problems, outperforming naive BC and IRL.

- We propose a theoretical method for confounded IL, purely from offline data, without expert queries nor exploration.

- We provide theoretical insight into the proposed methods by proving under strong assumptions that the expert's policy can be identified.

## 2 Related work

**Imitation learning**  Learning from demonstration has a long history with applications in autonomous driving (Pomerleau, 1988; Lu et al., 2022) and robotics (Schaal, 1999; Padalkar et al., 2023). Standard algorithms include BC, IRL (Russell, 1998; Ng et al., 2000; Ziebart et al., 2008), and adversarial methods (Ho & Ermon, 2016; Fu et al., 2017).

Imitation learning can suffer from a mismatch between the distributions faced by the expert and imitator due to the accumulation of errors when rolling out the imitator policy. This is commonly addressed by querying experts during the training (Ross et al., 2011) or by noise insertion in the expert actions Laskey et al. (2017).

Note that this issue is qualitatively different from the one we discuss in the paper: it is a consequence of the limited support of the expert actions and occurs even in the absence of the latent confounders.

Kumar et al. (2021) and Peng et al. (2020) consider a similar setting to ours where the environment has a latent variable, which explains the dynamics. They use privileged information to learn the dynamics encoder via supervised learning. In contrast, only the expert has access to privileged information in our setting.

**Causality-aware imitation learning**  Ortega & Braun (2010a;b) and Ortega et al. (2021) pointed out the issue of latent confounding and causal delusions that we discuss in this paper. In particular, Ortega et al. (2021) propose a training algorithm that learns the correct interventional policy. Unlike our algorithm, their approach requires querying experts during the training. However, as we discuss in section 4, their solution has weaker assumptions and also applies to non-Markovian dynamics.

Most similar to our work is Swamy et al. (2022b), which finds theoretical bounds on the gap between expert behavior and an imitator in the confounded setting, when imitating via BC, DAgger (Ross et al., 2011), which requires expert queries, or inverse RL. Inverse RL suffers from two key challenges: (1) it requires reinforcement learning in an inner loop with a non-stationary reward, and (2) the reward is typically learned via potentially unstable adversarial methods (Ho & Ermon, 2016; Fu et al., 2017). In contrast, our method trains the behavior policy using a well-behaved BC objective, which often enjoys better robustness and scalability compared to inverse RL.

There are various other works on the intersection of causality and IL which differ in setup. De Haan et al. (2019) consider the confusion of an imitator when the expert's decisions can be explained from causal and non-causal features of the state. It differs from our work as they assume the state to be fully observed, meaning it does not apply to our situation in which there is a latent confounder which needs to be inferred. This problem has been also discussed in Codevilla et al. (2019); Wen et al. (2020; 2022) and Spencer et al. (2021) under various names. Rezende et al. (2020) point out that the same problem appears in partial models that only use a subset of the state and find a minimal set of variables that avoid confounding. Swamy et al. (2022a) consider imitation learning with latent variables that affect the expert policy, but not the state dynamics, which is different from our case, in which the latent affects both the state and the expert's actions. Kumor et al. (2021) study the case in which a graphical model of the partially observed state is known and find which variables can be adjusted for so that conditional BC is optimal. An extension, Ruan et al. (2022), also handles sub-optimal experts.

**Meta-learning behaviors**  Our problem is related to meta-IL (Duan et al., 2017; Beck et al., 2023) where the aim is to train an imitation learning agent that can adapt to new demonstration data efficiently. Differently from our problem, the tasks in meta-IL can vary in the reward function. For imitation learning to work with the new reward functions, demonstrations of policies maximizing the new rewards are required. While meta-IL also considers a distribution of MDPs, the motivations and methods are different from ours. Our work does not consider adapting to new demonstrations, whereas meta-IL does not consider the confounding problem in the demonstrations.

Furthermore, our problem is related to meta-reinforcement learning (RL) (Duan et al., 2016; Wang et al., 2016; Beck et al., 2023), where an adaptive agent is trained to learn new tasks quickly via RL. Rakelly et al. (2019) and Zintgraf et al. (2020) propose meta-RL algorithms that consist of a task encoder and a task-conditional policy, similar to our inference model and latent-conditional policy. Zhou et al. (2019) propose agents that learn to probe the environment to determine the latent variables explaining the dynamics. Differently from our problem, the true reward function of the task is known.

## 3 Background

We begin by introducing confounded imitation learning. Following Ortega et al. (2021), we discuss how BC fails in the presence of latent confounders. We then define the interventional policy, the ideal (but a priori intractable) solution to the confounding problem.

### 3.1 Imitation learning

Imitation learning learns a policy from a dataset of expert demonstrations via supervised learning. The expert is a policy that acts in a (reward-free) Markov decision process (MDP) defined by a tuple $\mathcal{M} = (\mathcal{S}, \mathcal{A}, P(s' \mid s, a), P(s_0))$, where $\mathcal{S}$ is the set of states, $\mathcal{A}$ is the set of actions, $P(s' \mid s, a)$ is the transition probability, and $P(s_0)$ is a distribution over initial states. The expert's interaction with the environment produces a trajectory $\tau = (s_0, a_0, \ldots, a_{T-1}, s_T)$. The expert may maximize the expectation over some reward function, but this is not necessary (and some tasks cannot be expressed through Markov rewards (Abel et al., 2021)). In the simplest form of imitation learning, a BC policy $\pi_\eta(a \mid s)$ parametrized by $\eta$ is learned by maximizing the likelihood of the expert data, i.e., minimizing the loss $-\sum_{s,a \in \mathcal{D}} \log \pi_\eta(a \mid s)$, where $\mathcal{D}$ is the dataset of state-action pairs collected by the expert's policy.

### 3.2 Confounded imitation learning

We now extend the imitation learning setup to allow for some latent variables $\theta \in \Theta$ that are observed by the expert, but not the imitator. We define a family of Markov decision processes as a latent space $\Theta$, a distribution $P(\theta)$, and for each $\theta \in \Theta$, a reward-free MDP $\mathcal{M}_\theta = (\mathcal{S}, \mathcal{A}, P(s' \mid s, a, \theta), P(s_0 \mid \theta))$. Here, we make the crucial assumption that the latent variable is constant in each trajectory. We assume there exists an expert policy $\pi_{\exp}(a \mid s, \theta)$ for each MDP. When it interacts with the environment, it generates the following distribution over trajectories $\tau$:

$$P_{\exp}(\tau \mid \theta) = P(s_0 \mid \theta) \prod_{t=0}^{T} P(s_{t+1} \mid s_t, a_t; \theta) \pi_{\exp}(a_t \mid s_t; \theta).$$

In this setting, the trajectories from the expert distributions are called confounded, because the states and actions have an common ancestor, the latent variable $\theta$. The imitator does not observe this latent variable. It may thus need to implicitly infer it from the past transitions, so we take it to be a non-Markovian policy $\pi_\eta(a_t \mid s_1, a_1, \ldots, s_t)$, parameterized by $\eta$. The imitator generates the following distribution over trajectories:

$$P_\eta(\tau \mid \theta) = P(s_0 \mid \theta) \prod_{t=0}^{T} P(s_{t+1} \mid s_t, a_t; \theta) \, \pi_\eta(a_t \mid s_0, a_0, \ldots, s_t).$$

The Bayesian networks associated to these distributions are shown in Figure 1.

The goal of imitation learning in this setting is to learn imitator parameters $\eta$ such that when the imitator is executed in the environment, the imitator agrees with the expert's decisions. This means we wish to maximise $\mathbb{E}_{\theta \sim P(\theta)} \mathbb{E}_{\tau \sim P_\eta(\tau; \theta)} [\sum_{s_t, a_t \in \tau} \log \pi_{\exp}(a_t \mid s_t, \theta)]$. If the expert solves some task (e.g. maximizes some reward function), this amounts to solving the same task. The latent variable $\theta$ stays fixed during the entire interaction of the agent with the environment.

### 3.3 Naive behavioral cloning

If we have access to a data set of expert demonstrations, we can use BC on the demonstrations to learn the maximum likelihood estimate of the expert's policy. At optimality, this learns the *conditional policy*:

$$\pi_{\text{cond}}(a_t \mid s_1, a_1, \ldots, s_t) = \mathop{\mathbb{E}}_{\theta \sim p_{\text{cond}}(\theta \mid \tau)} \pi_{\exp}(a_t \mid s_t, \theta), \tag{1}$$

$$p_{\text{cond}}(\theta \mid \tau) \propto p(\theta) \prod_t p(s_{t+1} \mid s_t, a_t, \theta) \pi_{\exp}(a_t \mid s_t, \theta).$$

Following Ortega et al. (2021), consider the following example of a confounded multi-armed bandit with $\mathcal{A} = \Theta = \{1, \ldots, 5\}$ and $\mathcal{S} = \{0, 1\}$:

$$p(\theta) = \frac{1}{5}, \quad \pi_{\exp}(a_t \mid s_t, \theta) = \begin{cases} \frac{6}{10} & \text{if } a_t = \theta \\ \frac{1}{10} & \text{if } a_t \neq \theta \end{cases}, \quad P(s_{t+1} = 1 \mid s_t, a_t, \theta) = \begin{cases} \frac{3}{4} & \text{if } a_t = \theta \\ \frac{1}{4} & \text{if } a_t \neq \theta. \end{cases} \tag{2}$$

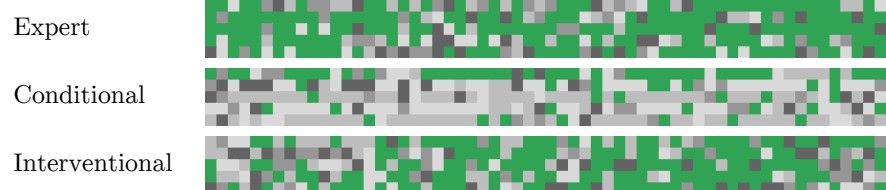

Figure 2: Actions from rollouts from bandit environment defined by Equation 2. The $x$-axis is episode time. In the $y$-axis five roll-outs are shown from the expert and policies Equation 1 and Equation 3. Colors denote actions, with the correct arm labelled green. The interventional imitator tends to the expert policy, while the conditional policy tends to repeat itself.

The expert knows which bandit arm is special (and labeled by $\theta$) and pulls it with high probability, while the imitating agent does not have access to this information. We define the reward in this bandit environment as $r_t = s_{t+1}$. However, note that we compare imitation learning algorithms that do not access the reward of the environment and only use the rewards for comparing the learned behaviors at evaluation time.

If we roll out the naive BC policy in this environment, shown in Figure 2, we see the causal delusion at work. At time $t$, inferring the latent by $p_{\text{cond}}$ takes past actions as evidence for the latent variable. This makes sense on the expert demonstrations, as the expert knows the latent variable. However, during an imitator roll-out, the past actions are not evidence of the latent, as the imitator is blind to it. Concretely, the imitator will take its first action uniformly and later tends to repeat that action, as it mistakenly takes the first action to be evidence for the latent.

## 3.4 Interventional policy

A solution to this issue is to only take as evidence the data that was actually informed by the latent, which is the dynamics distribution $p(s_{t+1} \mid s_t, a_t, \theta)$. This defines the following imitator policy:

$$\pi_{\text{int}}(a_t \mid s_1, a_1, \ldots, s_t) = \mathop{\mathbb{E}}_{\theta \sim p_{\text{int}}(\theta|\tau)} \pi_{\text{exp}}(a_t \mid s_t, \theta), \quad p_{\text{int}}(\theta \mid \tau) \propto p(\theta) \prod_t p(s_{t+1} \mid s_t, a_t, \theta) \tag{3}$$

In a causal framework, this corresponds to treating the choice of past actions as interventions. To see this, consider the distribution $p(a_t, s_1, \ldots, s_t | do(a_1, \ldots, a_{t-1}))$. By classic rules of the do-calculus (Pearl, 2009), this is equal to

$$\int_\Theta p(\theta) p(a_t | s_t, \theta) \prod_{\tau < t} p(s_{\tau+1} | s_\tau, a_\tau, \theta) d\theta$$

Taking the conditional $p(a_t | s_1, \ldots, s_t, do(a_1, \ldots, a_{t-1}))$ of that expression leads to equation (3). The policy in Equation 3 is therefore known as *interventional policy* (Ortega et al., 2021).

## 3.5 Failure of the conditional policy

Whether the conditional policy of Equation 1 is able to imitate expert behaviors under latent confounding depends on the relative contribution of evidence due to the dynamics $p(s_{t+1} \mid s_t, a_t, \theta)$ and the policy $\pi_{\text{exp}}(a_t \mid s_t, \theta)$ when inferring the latent in the posterior $p_{\text{cond}}(\theta \mid \tau)$. If most evidence comes from the dynamics, we expect the conditional and the interventional policies to be similar. On the other hand, when most evidence comes from the policy decisions, we expect large delusion. This is for instance the case when the expert policy is deterministic and the dynamics stochastic, or when the latent influences the state transitions only in parts of the state space, but always affects the expert behavior. In Appendix Appendix A, we quantitatively study the behavior of the conditional and interventional policies on the bandit example for varying amounts of dynamics and expert stochasticity.

### 3.6 Optimality of the interventional policy

To understand why the interventional policy may tend to the expert's policy consider $p_{int}(\theta|\tau)$ in Equation 3. Applying it to the bandit example, each subsequent observation $s_{t+1}$ after taking action $a_t$ provides some evidence for the latent $\theta$: if we find $s_{t+1} = 1$, then the latent $\theta$ is more likely to equal $a_t$, if we find $s_{t+1} = 0$, then $\theta$ is less likely to equal $a_t$. In Figure 2, we see that this "true interventional" indeed approaches the expert's policy. This is further illustrated in subsection 6.1. In this case, the interventional policy thus presents a solution to the confounding problem. In the rest of this paper, we focus on the question if and how it can be learned from data.

Note that the interventional policy may only achieve optimal performance after observing many time steps. However, it is not guaranteed to be the policy that adapts identifies the latent as fast as possible. This may require a form of active exploration (Zhou et al., 2019) that is beyond the scope of this work.

## 4 Deconfounding imitation learning

In this section, we discuss how, instead of the conditional policy, we can learn the interventional policy and thus fix the confounding problem. This improvement we call deconfounding the imitation learning. First, we illustrate how this is provably possible in theory under strong assumptions. We then show how to learn it in practice, even when these assumptions don't fully hold.

### 4.1 Identifying the interventional policy in theory

Based on the graphical model shown in Figure 1, and applying the rules of do-calculus, we can identify the interventional policy $\pi_{int}(a_t \mid s_1, a_1, \ldots, s_t) = p(a_t|s_1, \ldots, s_t, do(a_1, \ldots, a_{t-1}))$ by back-door adjustment Pearl (2009) if we observe the latent variable $\theta$. This results in Equation 3. But since the latent $\theta$ is unobserved, the interventional policy is not identifiable without further assumptions on the distribution.

However, we will now show that under some assumptions, the interventional policy does become identifiable. First, we assume that the latent variable $\theta$ of the MDP is static, or in other words stays fixed during the interaction of the policy with the MDP. This makes the dynamics and expert policy stationary in each trajectory. If we combine this with the strong assumption of *recurrence*, commonly used in theoretical reinforcement learning (Watkins & Dayan, 1992, Thm. 1), the interventional policy becomes identifiable.

**Assumption 1.** *We assume that the MDP is* recurrent *(Norris, 1997, Sec. 1.5), meaning that all state-action pairs $s, a$ are reached infinitely often in each trajectory.*

If the MDP is recurrent with the expert policy, it is possible to identify the interventional policy in theory from an infinite dataset $\mathcal{D}$ of expert demonstrations of infinite length. On each individual trajectory $\tau_i$ with index $i$, we are able to count the state and action transitions to estimate the transition probability $\hat{p}_i(s'|s, a)$ and expert policy $\hat{\pi}_i(a|s)$ of that trajectory. We can then compute the likelihood of a dataset trajectory $i$ given a state-action sequence $(s_1, a_1, \ldots, s_t)$ based on the estimated environment transitions, $\prod_t \hat{p}_i(s_{t+1}|s_t, a_t)$. The corresponding belief over dataset trajectories $i$ follows from Bayes' theorem as

$$\hat{p}_{int}(i|s_1, a_t, ..., s_t) = \frac{\prod_t \hat{p}_i(s_{t+1}|s_t, a_t)}{\sum_{i'} \prod_t \hat{p}_{i'}(s_{t+1}|s_t, a_t)} .$$

Then we can estimate

$$\hat{\pi}_{int}(a_t|s_1, a_1, .., s_t) = \mathbb{E}_{i \sim \hat{p}_{int}(i|s_1, a_t, ..., s_t)} \hat{\pi}_i(a_t|s_t) . \tag{4}$$

This successfully identifies the interventional policy:

**Theorem 1.** *From infinitely many demonstrations of infinite length from a MDP that is recurrent with the expert policy $\pi_{exp}$, we have that $\hat{\pi}_{int}$ from Equation 4 equals $\pi_{int}$ from Equation 3.*

*Proof.* Because of the recurrence assumption, we can correctly estimate $\hat{p}_i(s'|s,a) = p(s'|s,a,\theta_i)$ and $\hat{\pi}_i(a|s) = \pi_{\exp}(a|s,\theta_i)$. Then, suppressing a normalization factor that is constant in $a_T$, we write:

$$\pi_{\mathrm{int}}(a_T|s_1,...,s_T) \propto \int_\Theta p(\theta) \left[ \prod_t p(s_{t+1}|s_t,a_t,\theta) \right] \pi_{\exp}(a_T|s_T,\theta) d\theta$$

$$\propto \int_\Theta \sum_i p(\theta|i)p(i) \left[ \prod_t p(s_{t+1}|s_t,a_t,\theta) \right] \pi_{\exp}(a_T|s_T,\theta) d\theta$$

$$\propto \sum_i p(i) \left[ \prod_t p(s_{t+1}|s_t,a_t,\theta_i) \right] \pi_{\exp}(a_T|s_T,\theta_i)$$

$$\propto \sum_i p(i) \left[ \prod_t \hat{p}_i(s_{t+1}|s_t,a_t) \right] \hat{\pi}_i(a_T|s_T)$$

$$\propto \hat{\pi}_{\mathrm{int}}(a_T|s_1,...,s_T)$$

where we recognized $p(\theta|i)$ as the (Dirac) delta distribution around $\theta_i$, and $p(i)$ is the uniform distribution over samples from the dataset $\mathcal{D}$. □

In this proof, the stationarity of the dynamics and expert make it possible to collect statistics across the time steps into distributions. Furthermore, the proof crucially relies on the recurrence assumption, as it allows us to evaluate the likelihood of any state-action sequence on each of the trajectories' distributions. Without this assumption, we will in general only be able to learn an approximation of the likelihood from state-action pairs observed in the trajectory.

The interventional policy estimated in Equation 4 requires marginalization over the entire demonstration dataset and thus isn't very practical. Instead, we propose to match the expert data with a learned latent variable model $p_{\psi,\eta}$

$$p_{\psi,\eta}(\tau) = \int_{\hat{\Theta}} p_\psi(\hat{\theta}) \prod_t p_\psi(s_{t+1}|s_t,a_t,\hat{\theta}) \pi_\eta(a_t|s_t,\hat{\theta}) d\hat{\theta}. \tag{5}$$

Similar to the argument in Theorem 1, under the same assumptions, if we match the ground-truth demonstration likelihood $p(\tau)$ with the latent variable model $p_{\psi,\eta}$, then the interventional policy derived from the latent variable model $p_{\psi,\eta}$ matches the ground truth interventional policy $\pi_{\mathrm{int}}$.

## 4.2 Learning from demonstrations and explorations

If each expert demonstration covers the entire state-action space, we have shown that we can estimate state dynamics and expert policy, and thus the interventional policy using Equation 4 or the latent variable model in Equation 5.

What if this strong assumption on the expert data does not hold? We can make the problem easier by allowing our agent to interact with the environment during training. That allows us to see more samples of the environment dynamics, making up for the lack of recurrence guarantees. Allowing interactions with the environment is a common modification to the imitation learning problem, used for example in IRL.

In that case, if we explore with policy $\pi_{\mathrm{expl}}(a|s)$, the state distribution is

$$p_{\mathrm{expl}}(\tau) = \int_\Theta p(\theta) \prod_t \pi_{\mathrm{expl}}(a_t|s_t) p(s_{t+1}|s_t,a_t,\theta) d\theta. \tag{6}$$

The corresponding latent model is given by

$$p_{\psi,\mathrm{expl}}(\tau) = \int_{\hat{\Theta}} p_\psi(\hat{\theta}) \prod_t \pi_{\mathrm{expl}}(a_t|s_t) p_\psi(s_{t+1}|s_t,a_t,\hat{\theta}) d\hat{\theta}, \tag{7}$$

where we assume that we can evaluate the likelihood of the exploration policy.

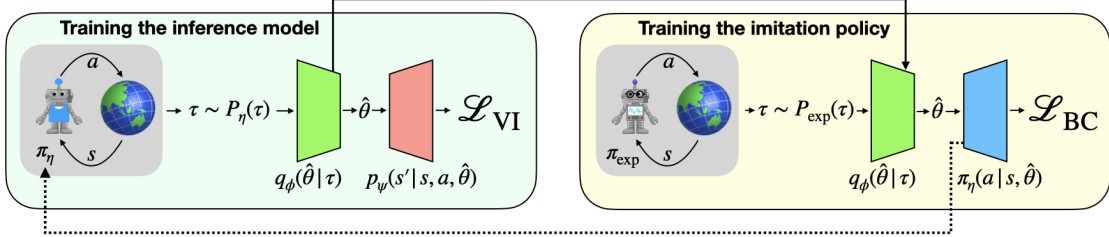

Figure 3: Overview of the method. On the left, the inference model consisting of the encoder $q_\phi$ and decoder $p_\psi$ is trained using variational inference. Training data is sampled from the environment using an exploratory policy. The dotted arrow depicts how the exploratory policy can optionally be the imitation policy $\pi_\eta$ trained on the right. On the right, the imitation policy is trained with behavioral cloning on expert data. The expert does not need to interact with the environment at training time. Instead, stored expert trajectories can be used. To learn the deconfounded policy, the encoder trained on the left is used for inferring the latent variable on the expert trajectories.

We propose to learn the latent model of Equation 7 via amortized variational inference, using an inference model $q_\phi(\hat{\theta}|\tau)$ (Kingma & Welling, 2014) that infers the latent variable from a trajectory. This involves maximizing the evidence lower bound (ELBO), a lower bound of the log likelihood, with respect to $\psi$ and $\phi$:

$$
\begin{aligned}
\mathbb{E}_{\tau \sim p_{\text{expl}}} \log p_{\psi,\text{expl}}(\tau) &\geq \mathbb{E}_{\tau \sim p_{\text{expl}}} \mathbb{E}_{\hat{\theta} \sim q_\phi(\hat{\theta}|\tau)} \left[ \log p_\psi(\hat{\theta}) - \log q_\phi(\hat{\theta}|\tau) + \sum_t \log \pi_{\text{expl}}(a_t|s_t) + \log p_\psi(s_{t+1}|s_t,a_t,\hat{\theta}) \right] \\
&= \mathbb{E}_{\tau \sim p_{\text{expl}}} \mathbb{E}_{\hat{\theta} \sim q_\phi(\hat{\theta}|\tau)} \left[ \log p_\psi(\hat{\theta}) - \log q_\phi(\hat{\theta}|\tau) + \sum_t \log p_\psi(s_{t+1}|s_t,a_t,\hat{\theta}) \right] + \text{const} = \mathcal{L}_{\text{VI}}
\end{aligned}
\tag{8}
$$

where we can ignore the exploration policy likelihood as it does not depend on the parameters. At optimality, and assuming sufficient expressivity, the inference model learns the model posterior

$$
q_\phi(\hat{\theta}|\tau) = p_{\psi,\text{expl}}(\hat{\theta}|\tau) \propto p_\psi(\hat{\theta}) \prod_t p_\psi(s_{t+1}|s_t,a_t,\hat{\theta}).
$$

Crucially, the learned inference model $q_\phi(\hat{\theta}|\tau)$ does not depend on the likelihood of the policy selecting actions. This is because we explore with a policy that—unlike the expert—does not depend on the latent $\theta$. Therefore, we learn exactly the interventional latent inference model from Equation 3, with the learned dynamics.

In a next step, we use the inference model $q_\phi(\hat{\theta}|\tau)$ together with a learned dynamics model $p_\psi(s_{t+1}|s_t,a_t,\hat{\theta})$ to learn an imitation policy $\pi_\eta(a|s,\hat{\theta})$, that approximates the expert policy $\pi_{\text{exp}}(a|s,\theta)$. Again, we use variational inference and train the model $p_{\psi,\eta}$ from Equation 5 with samples from the demonstrations. Thus, we maximize $\eta$ in the ELBO

$$
\mathbb{E}_{\tau \sim p} \log p_{\psi,\eta}(\tau) \geq \mathbb{E}_{\tau \sim p} \mathbb{E}_{\hat{\theta} \sim q_\phi(\hat{\theta}|\tau)} \left[ \log \pi_\eta(a_t|s_t,\hat{\theta}) \right] + \text{const} = \mathcal{L}_{\text{BC}},
\tag{9}
$$

which effectively becomes a behavioral cloning loss. Here we omitted terms constant in the policy parameters $\eta$.

The training based on these two objectives is illustrated in Figure 3. By optimizing the loss in Equation 8, we learn a model to infer the latent variable just based on the dynamics of a trajectory; by optimizing the loss in Equation 9, we learn a policy conditional on that latent. Combined, we learn to approximate the interventional policy of Equation 3.

# 5  Practical algorithm

We now present a practical algorithm for training an agent from expert data in the presence of latent confounders. As outlined in section 4, we learn to infer the latent variable by interacting with the environment using an exploratory policy, and learn to clone the expert on the demonstrations conditioned on the inferred latent. At test time, the agent alternates between updating its belief about the latent variable and imitating an expert dictated by its current belief. In Appendix C, we describe an algorithm that does not require the ability to gather more data interactively, but faces a more difficult learning problem in practice.

**Components**   The agent consists of: 1) an inference model $q_\phi$, which maps trajectories $\tau = (s_0, a_0, \ldots, s_n)$ to a belief over a latent variable $q_\phi(\hat{\theta} \mid \tau)$; 2) a dynamics model $p_\psi$ mapping latent, state, and action to a distribution over the next state $p_\psi(s' \mid s, a, \hat{\theta})$; and 3) a latent-conditional policy $\pi_\eta(a \mid s, \hat{\theta})$. An overview of the training algorithm is presented in Figure 3.

**Training the model for online inference**   The inference model training closely follows the outline given in section 4. However, since at test time, the agent needs to infer the latent online from partial trajectories, we adjust the model and the ELBO of Equation 8 slightly. At timestep $t$, the encoder $q_\phi$ takes as input the trajectory observed until timestep $t$, and predicts a distribution of the belief over the latent $\hat{\theta}$.

We follow Zintgraf et al. (2020) by defining the prior at timestep $t$ as the inferred latent distribution from timestep $t - 1$, starting with a diagonal Gaussian with unit variance as the initial prior. This process is similar to Bayesian filtering, where beliefs about the state of a process are updated sequentially in response to new evidence. As such, each prior evolves based on the latest observed data, similar to the updating mechanisms in Kalman filters and other Bayesian state estimators (Kalman, 1960; Krishnan et al., 2015). The modified ELBO can then be written as

$$\hat{\mathcal{L}}_{\mathrm{VI}} = \mathbb{E}_{\tau \sim p_{\mathrm{expl}}} \left[ \mathbb{E}_{\hat{\theta} \sim q_\phi(\hat{\theta}|\tau_{:t})} \left[ \log p_\psi(s_{t+1} \mid s_t, a_t, \hat{\theta}) \right] - \beta D_{KL}\Big( q_\phi(\hat{\theta} \mid \tau_{:t}) \,\Big\|\, q_\phi(\hat{\theta} \mid \tau_{:t-1}) \Big) \right], \qquad (10)$$

where $D_{KL}$ is the KL divergence and $\tau_{:t}$ denotes the trajectory until timestep $t$. Following prior work on VAEs, we use a coefficient $\beta$ for the prior regularization (Higgins et al., 2017).

As an exploration policy, we use the latent-conditional policy $\pi_\eta$ conditioned on $\hat{\theta}$ inferred by $q_\phi$. This is a convenient choice because using $\pi_\eta$ means we do not have to train multiple policies. Furthermore, using $\pi_\eta$ for exploration biases the training data distribution for the inference model toward data that the policy encounters when it is deployed in the environment potentially improving the generalization of the inference model. In practice, we condition the policy on the mean of the inferred distribution instead of a sample from it. We find that using either does not make a big difference. Other exploration policies may be used as long as they explore sufficiently diverse trajectories and do not depend on the true latent $\theta$.

As described in section 4, the learned inference model is used for inferring the latents on the expert trajectories $\tau_e^j$. The policy $\pi_\eta$ is then trained to minimize Equation 9. We show the pseudocode for the full training algorithm in Appendix B.

**Test time**   At test time, the agent faces an environment with an unknown latent and needs to adapt to the correct expert behavior. We solve this problem by alternating between updating a posterior belief over the latent and acting under the current belief. Concretely, the agent initially samples a latent from the prior $\hat{\theta} \sim \mathcal{N}(0, \mathbb{1})$ and an action $a \sim \pi_\eta(a|s, \hat{\theta})$ to imitate the expert corresponding to that latent. It observes the state transition and computes the posterior belief with the inference network. Another latent is sampled from the updated belief, and so on. In practice, as during exploration, we do not sample from the inferred distribution but instead condition the policy on the mean. Once the inference has converged to match the true latent for the environment, the true expert for the environment will be imitated consistently. We summarize the test-time behavior in pseudocode in Appendix B.

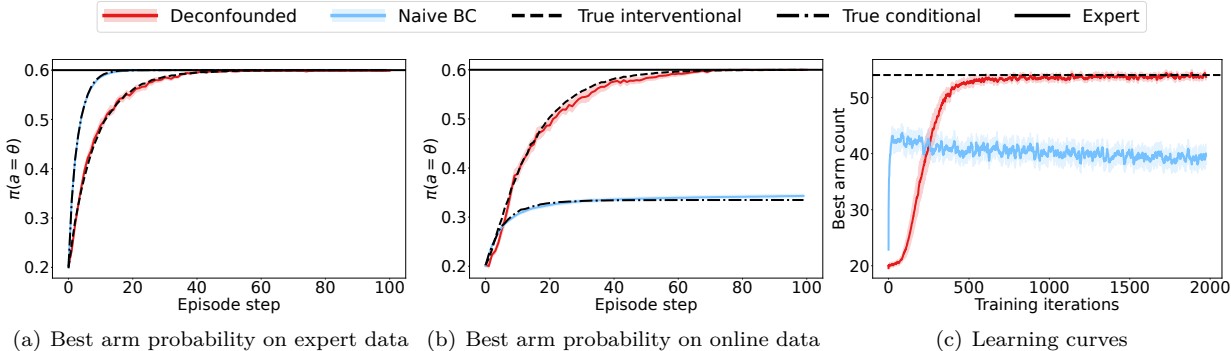

(a) Best arm probability on expert data    (b) Best arm probability on online data    (c) Learning curves

Figure 4: Imitation learning in a multi-armed bandit problem. The shading shows the standard error of the mean. The left panel compares the policies when evaluated on trajectories sampled by the expert policy. The x-axis is the step on the trajectory and the y-axis is the probability of choosing the best arm. The middle panel is otherwise the same, except run on trajectories sampled online with the policies themselves. The right panel shows the learning curves. The x-axis shows training iterations and the y-axis shows the number of times the best arm is chosen by the policy under training during a trajectory with 100 time steps. The curves are averages for sliding window of length 10 training iterations.

## 6 Experiments

To test our method in practice, we conduct experiments in the multi-armed bandit problem from Ortega et al. (2021) and in multiple control environments. We aim to answer three questions: 1) How big is the effect of confounding on naive BC — large enough to justify the use of specialized methods? 2) Is our algorithm capable of identifying the interventional policy? 3) How well does the interventional policy imitate the expert?

### 6.1 Investigating deconfounding in a multi-armed bandit

We begin the empirical study by experimenting with the multi-armed bandit problem proposed by Ortega et al. (2021) and described in section 3. The expert policy is defined in Equation 2. We consider episodes of length 100. As we are only interested in the effects of confounding on imitation learning, and not in effects arising from over-fitting a small dataset, we generate new training data from the expert for each update of the learning algorithms. Each learning algorithm is run for ten independent seeds and the results are averaged. The hyperparameters for the algorithms are provided in Appendix D.

**Naive BC and the conditional policy**    To answer our first question, we compare a naive BC to the true conditional policy described in section 3. As the latter policy is non-Markovian, we also allow the BC policy to observe the history. One way to enable this adaptation is to equip the agent with a memory, which the agent can learn to use for representing its belief about the latent variable. To provide such a memory, we implement the imitation learner as a recurrent neural network (RNN). Figure 4 a) shows the probability the different policies assign to choosing the best arm when evaluated on data collected by the expert policy. We see that the naive BC agent learns a policy that matches the true conditional policy closely. This results in problems for the policy learned with naive BC when it is deployed in the environment and has to choose the actions itself, as shown in Figure 4 b). The naive BC closely tracks the action probability of the true conditional policy, which performs much worse than the expert policy. These results suggests that it has suffered the full impact of the confounding problem.

**Deconfounded imitation learning and the interventional policy**    To answer our second question, we implement the deconfounded imitation learner as described in section 5.Figure 4 a) and b) show that the proposed method closely matches the true interventional policy both on expert trajectories and online. Figure 4 c) shows the number of times the policies chose the best arm during an episode. Our deconfounded

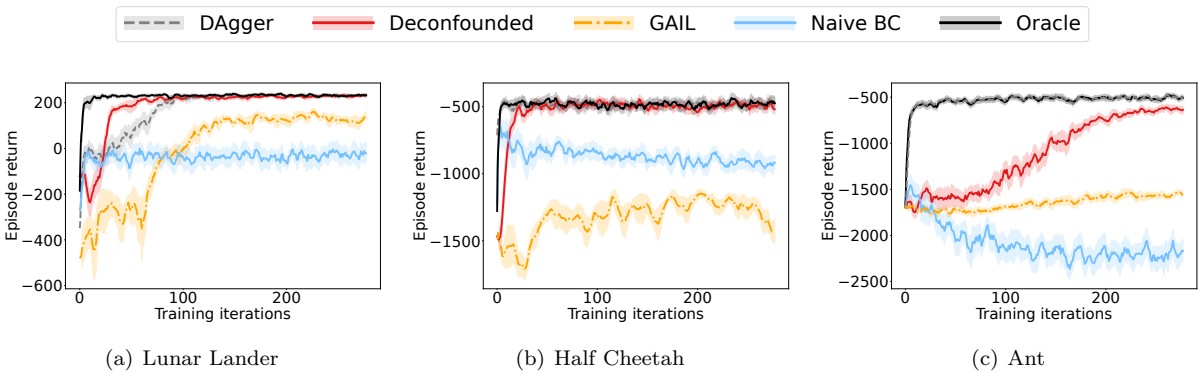

Figure 5: Experiments in our confounded, stochastic environments. We show the episodic return of each agent over the course of training. The curves for the are averages sliding window of length 5. The shading shows the standard error of the mean.

imitator converges to the true interventional policy, showing that it learned to optimally imitate the expert in the presence of latent confounders, and answering our third question. This near-perfect imitation performance comes at the price of requiring exploration data to train the inference model as well as an increased number of training iterations needed for convergence.

## 6.2 Demonstrating deconfounding in confounded MDPs

Next, we demonstrate that the confounding issue affects MDPs with considerably more complex dynamics than the bandit and that the answers to the three questions do not change with the increased complexity. For `LunarLander-v2` (Brockman et al., 2016), we consider a modified version with unknown key bindings: a latent $\theta$ specifies a permutation in the map between two of the the agent's actions and the behaviors (firing the left and right engines of the space craft). This permutation is known to the expert, but not the imitator. For `HalfCheetahBulletEnv-v0` (Coumans & Bai, 2016–2021), we modify the environment similarly to Swamy et al. (2022b) by varying the target speed. The expert observes the true target speed while the imitator only observes a noisy indicator, which shows whether it is running faster or slower than the target speed. For `AntGoal-v0` (Todorov et al., 2012), we consider a version, where the task is to run to a goal randomly sampled from within a circle around the starting point. The expert observes the true goal coordinates, while the imitator only observes a noisy indicator of the goal direction and distance. In all environments, in addition to providing the imitator with less information about the latent variable than the expert, we make the environment somewhat stochastic. These changes make the naive imitators try to infer the latent from the more deterministic expert actions rather than the stochastic dynamics, which results in the confounding problem. For full details about the environments, and how the confounding problem may arise in each of them, please see the Appendix E.

The expert policies are trained with proximal policy optimization (Schulman et al., 2017). In order to avoid finite-sample-size effects, we use an infinite-size training dataset by generating expert trajectories on the fly. Each learning algorithm is run for 5 or 6 independent seeds and the results are averaged. Additional details are provided in Appendix E.

**Naive BC** Like in the bandit example, we first analyze how strongly confounding affects the naive BC policy. The naive learner is again implemented as an RNN. In Figure 5 we see that the naive BC agent fails to imitate the expert behavior and performs much worse than an oracle imitator, which is otherwise exactly the same setting but it is trained with knowledge of the latents. This failure to generalize is again evidence for causal delusions: the agent learned to infer the latent from the expert behavior rather than the noisy dynamics of the environment.

**Deconfounded imitation learning** To solve the confounding issue in these environments, we first test the DAgger algorithm (Ross et al., 2011), which queries the experts during training. We find that it indeed solves the confounding problem: the agent quickly approaches the performance of the oracle imitator in all three environments.

The DAgger performance serves as an unachievable upper bound for our method, as it not only solves the confounding issue but also reduces the more common distribution shift issue present in imitation learning. However, recall that the expert policy may be defined by, e.g., a human expert, who would be expensive or impossible to query at imitation learning time making DAgger a potentially difficult method to use in practice.

Can our deconfounded imitator also solve the confounding problem? We test the deconfounded imitation learning algorithm described in section 5 on the control environments. We find it beneficial to modify the described algorithm in two ways: 1) extending the reconstruction loss in Equation 10 to multi-step predictions (Hafner et al., 2020); 2) conditioning the policy on the inferred latent $\hat{\theta}_t^{\hat{j}}$ at the current timestep $t$ instead of the last time step $\hat{\theta}_H^j$ when training the policy. See Appendix E for details.

In Figure 5, we see that the deconfounded learner clearly outperforms the naive BC baseline, matching the performance of the policy learned by DAgger in two out of three environments and coming close in the third. While it is hard to interpret what functions high-dimensional neural networks have learned exactly, matching the performance of DAgger, which we know can recover the interventional policy, is encouraging. It suggests that the deconfounded agent learned to infer the latent from environment transitions and learn a policy that acts like the expert under the inference. In the Ant environment, it is impossible to infer the direction from the noisy indicator from a single observation. Therefore, the agent may start moving the wrong way in the beginning of the episode. Recovering from such a false start may be much easier with the help of extra supervision from the expert, giving DAgger a particularly strong advantage in this environment.

**Inverse reinforcement learning** In theory, IRL is capable of recovering the expert performance even in confounded environments (Swamy et al., 2022b). We test this theory in practice, by comparing to generative adversarial imitation learning (GAIL) (Ho & Ermon, 2016). We implemented GAIL closely following a popular publicly available implementation[1] and using recurrent PPO by Raffin et al. (2021) as the RL algorithm. To stabilize GAIL, we sample four times more data for every training step compared to the other algorithms.

In LunarLander, GAIL is able to achieve higher returns than Naive BC, but does not recover the performance of the proposed method. In HalfCheetah and Ant, GAIL does not recover the performance of Naive BC. In Appendix D, we provide results for GAIL in the bandit environment from subsection 6.1. We found that in the bandit, GAIL does not recover the interventional policy and therefore does not converge to the expert behavior during an episode. This is because the reward function defined by GAIL can always be maximized by a deterministic policy, even in the case when the expert is a stochastic policy, like in the bandit. From these results, we conclude that while IRL is in theory capable of solving confounded imitation learning problems, where the expert is a deterministic policy, getting it to work well can be difficult in practice.

## 7 Limitations

While the proposed method recovers the policy DAgger learns under ideal conditions, it does not address the original challenge DAgger is designed to solve. That is, if the expert data does not cover the state-action space sufficiently, the proposed method may not be enough to learn the optimal policy. Furthermore, while the proposed method does not require access to the expert, it does require sampling access to the environment similarly to IRL. To find the deconfounded policy fully offline, an algorithm outlined in the in section 4 could be used. For the methods to work provably, we need to make strong assumptions of recurrence.

Following Ortega et al. (2021) we model the confounded IL setting with a CMDP where the latent stays fixed during the entire interaction of the policy with the environment. This may be a limiting assumption,

---

[1] https://github.com/HumanCompatibleAI/imitation

for example in a driving scenario, where the environment may have latent factors that govern the dynamics over a small section of the road but the agent is expected to be able to traverse many different sections.

Finally, while the environments we experiment in are commonly used to evaluate RL and IL policies, we acknowledge that these are fairly limited domains and do not reflect the full complexity of training useful policies for the real world.

The components of our algorithm, variational inference and behavioral cloning, are commonly used techniques elsewhere in machine learning. Therefore, scaling the proposed method to real world problems should be possible. However, we did not optimize for sample complexity of either learning algorithm, which would limit the application of our method to real world problems. Developing a more sample efficient version of the algorithm and testing it in real world domains is an exciting avenue of future work.

## 8 Conclusion

Naive imitation learning algorithms can fail in the presence of latent confounders — for instance when the expert has access to more information than the imitator. This work presents a breakdown of this confounding problem. First, we studied under which conditions latent confounding impacts the performance of BC. We demonstrated that this issue is more relevant when there is substantial stochasticity in environment transitions or the expert policy is nearly deterministic. We then analyzed in which settings the interventional policy, which solves the confounding issue, is identifiable without query access to the expert.

Informed by the theoretical results, we proposed a practical algorithm for deconfounding imitation learning with variational inference that provably converges to the interventional policy. While this problem has been previously studied in theory with inverse RL, we propose a novel variational inference solution, which we analyze theoretically and evaluate in practice. Finally, we evaluated the proposed method with experiments in a multi-armed bandit and confounded MDPs. We found it was able to learn interventional policies in all of the settings, alleviating the confounding problem that limits naive imitation learning.

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

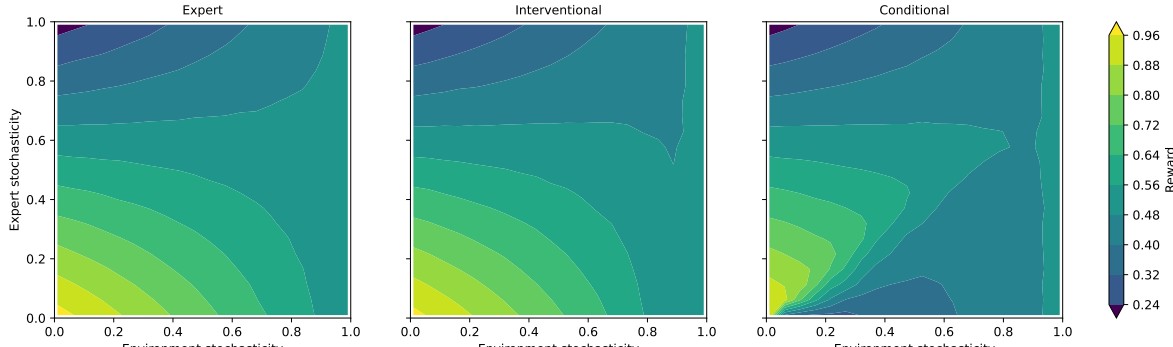

Figure 6: Conditional and interventional policies in the bandit environment Equation 2 as a function of environment stochasticity and expert stochasticity. We show the expected success probability for the expert (left), the interventional policy (middle), and the conditional policy (right). Along the axes, we vary the expert policy from deterministic (bottom) to stochastic (top) and the environment transitions from deterministic (left) to stochastic (right); see text for details. The interventional policy performs similar to the expert, while the conditional policy performs worse due to latent confounding when the environment stochasticity is larger than the expert stochasticity.

## A  Confounding and stochasticity

The behavior success of the conditional and interventional policies depend on the stochasticity of the environment and the expert, as this affects the evidence for the inference of the latent variable. If most evidence comes from the dynamics, which would e. g. be the case when the expert policy is very noisy and the dynamics deterministic, we expect the conditional and the interventional policies to be similar. However, when most evidence comes from the policy decisions, we expect large delusion.

See figure Figure 6 for a plot of the rewards of the various policies given varying amounts of stochasticity. Instead of the fixed coefficients of Equation 2, we vary $\pi_{\exp}(a_t = \theta)$ between 1 (deterministic expert) and $\frac{1}{5}$ (maximally stochastic expert) and $P(s_{t+1} = 1 | a_t = \theta)$ between 1 (deterministic environment) and $\frac{1}{2}$ (maximally stochastic environment).

In the left panel, we show the expected success probability of the expert policy: as expected, it is highest when both the expert and the environment are deterministic. The interventional policy of Equation 3 performs similarly. However, the conditional policy, shown in the right panel, is only able to match the expert behavior if the stochasticity in the expert policy outweighs the stochasticity in the environment. When the environment is stochastic and the expert comparably deterministic, the conditional policy (and thus naive BC) infers latents largely from past actions, suffers from delusions, and fails to imitate.

## B  Training deconfounded imitators

Algorithm 1 shows how an agent is trained in practice from expert data in the presence of latent confounders. Algorithm 2 shows the test time implementation of the deconfounded imitation learners, which alternate between updating a posterior belief over the latent and acting under the current belief.

## C  Deconfounded imitation learning from expert data alone

As discussed in section 4, when suitable demonstrations from the expert are available, the interventional policy is identifiable from the expert data alone, without access to the environment or the expert. In this section, we present a practical algorithm for learning the deconfounded imitation policy from such an expert dataset. At test time, the agent works the same way as the Algorithm 1 presented in the paper.

**Data:** Initial parameters of the imitation policy $\eta$, inference model $\phi$, and dynamics model $\psi$; expert dataset $\{\tau_e^j\}$, MDP family $\mathcal{M}_\theta$, true latent distribution $p(\theta)$, learning rates $\alpha_1$, $\alpha_2$, $\alpha_3$.

**while** *not done* **do**

    $\theta \sim p(\theta)$, $s_0 \sim p_0(s_0)$;

    $\tau = s_0$, $t = 0$;

    **for** $t \leq H$ **do**

        $\hat{\theta}_t \sim q_\phi(\hat{\theta}_t \mid \tau_{:t})$;                  `// Infer the latent for trajectory`

        $a_t \sim \pi_\eta(a_t \mid s_t, \hat{\theta})$;                `// Sample action from a Markov policy`

        $s_{t+1} \sim p(s_{t+1} \mid s_t, a_t, \theta)$;                `// True dynamics`

        Append $(a_t, s_{t+1})$ to $\tau$;

        $t \leftarrow t + 1$;

    **end**

    $\phi \leftarrow \phi - \alpha_1 \nabla_\phi \hat{\mathcal{L}}_{\text{VI}}(\tau)$;              `// Sample estimate of Equation 10`

    $\psi \leftarrow \psi - \alpha_2 \nabla_\psi \hat{\mathcal{L}}_{\text{VI}}(\tau)$;

    $\hat{\theta}_H^j \sim q_\phi(\hat{\theta}_H^j \mid \tau_e^j)$;           `// Infer latent from $j$-th expert trajectory`

    $\eta \leftarrow \eta - \alpha_3 \nabla_\eta \sum_j \sum_{s,a \in \tau_e^j} \log \pi_\eta(a \mid s, \hat{\theta}_H^j)$;

**end**

**Algorithm 1:** Training deconfounded imitators

**Data:** Parameters of the policy $\eta$ and inference model $\phi$; MDP $\mathcal{M}_\theta$ with ground-truth latent $\theta$.

$s_0 \sim p_0(s_0)$;

$\tau \leftarrow s_0$, $t \leftarrow 0$;

**for** $t \leq H$ **do**

    $\hat{\theta}_t \sim q_\phi(\hat{\theta}_t \mid \tau_{:t})$ ;                  `// Infer the latent for trajectory`

    $a_t \sim \pi_\eta(a_t \mid s_t, \hat{\theta}_t)$ ;                 `// Condition on inferred latent`

    $s_{t+1} \sim p(s_{t+1} \mid s_t, a_t, \theta)$ ;                `// True dynamics`

    Append $(a_t, s_{t+1})$ to $\tau$;

    $t \leftarrow t + 1$;

**end**

**Algorithm 2:** Deconfounded imitators at test time

Training an inference model directly on the expert demonstration faces the same problem as naive imitation learning, i.e., the trained model takes the expert's actions as evidence for the latent. However, by the assumption, that the expert is uniquely determined by the environment dynamics, we can directly learn the conditional policy and dynamics model explaining the expert trajectories from the demonstrations because a latent that explains the dynamics also explains the expert. To train such models, we use variational inference to learn a trajectory encoder $q_{\phi_{\text{off}}}$, which infers the latent for the expert trajectories, and a factorized decoder, which reconstructs the dynamics of the environment and the expert's policy using networks $p_\psi$ and $\pi_\eta$ respectively. The variational inference objective is given by

$$\mathcal{L}_{\text{off},i} = \mathbb{E}_{\hat{\theta} \sim q_\phi(\hat{\theta}|\tau_e^i)}\left[\sum_{t=0}^{H} \log p_\psi(s_{t+1} \mid s_t, a_t, \hat{\theta}) + \log \pi_\eta(a_t|s_t, \hat{\theta})\right] - \beta D_{KL}\Big(q_{\phi_{\text{off}}}(\hat{\theta} \mid \tau_e^i) \,\Big\|\, p(\hat{\theta})\Big), \qquad (11)$$

which, unlike the objective in the main paper, represents a VAE where the encoder $q_{\phi_{\text{off}}}$ takes as input the full expert trajectory $\tau_e^i$, and the decoder decodes both the action and transition probabilities throughout the trajectory.

This gives us a way for training the conditional policy imitating the experts in the demonstrations and a dynamics model. However, we cannot directly use the learned inference model $q_{\phi_{\text{off}}}$ for implementing the interventional policy, because it takes the expert's actions as evidence for the latent. The Algorithm 1 works by separately learning an inference model from interactions with the environment and using that inference model for deconfounding the expert trajectories. When we do not have sampling access to the environment, we cannot learn the inference model directly. Instead, we observe that one factor of the decoder used for training the inference model is a dynamics model of the environment conditional on the predicted latent. Therefore, we can use it to generate synthetic trajectories for training an online inference model $q_{\phi_{\text{on}}}$ to minimize the online variational inference objective given in the main paper. The online inference model can then be used for implementing the interventional policy similarly as in Algorithm 2. The full offline dynamics learning algorithm is presented in Algorithm 3.

**Data:** The initial parameters of the imitation policy $\eta$, offline inference model $\phi_{\text{off}}$, online inference model $\phi_{\text{on}}$, dynamics model $\psi$, prior distribution for the learned latent space $p(\tilde{\theta})$, a dataset of expert trajectories $\{\tau_e^i\}$, an MDP $(\mathcal{S}, \mathcal{A}, p, p_0, H)$, learning rates $\alpha_1$, $\alpha_2$, $\alpha_3$, and $\alpha_4$.

**while** *not done* **do**

$\quad \tilde{\theta} \sim p(\tilde{\theta})$ ;            `// Sample from the prior`

$\quad s_0 \sim p_0(s_0)$ ;           `// Sample from a learned model or expert data`

$\quad \tau_{\text{synth}} = s_0, t = 0$;

$\quad$ **for** $t \leq H$ **do**

$\quad\quad a_t \sim \pi(a_t \mid s_t)$ ;          `// Sample action from a Markov policy`

$\quad\quad s_{t+1} = p_\psi(s_{t+1} \mid s_t, a_t, \tilde{\theta})$ ;        `// Dynamics model`

$\quad\quad$ Append $(a_t, s_{t+1})$ to $\tau_{\text{synth}}$;

$\quad\quad t = t + 1$;

$\quad$ **end**

$\quad \phi_{\text{on}} = \phi_{\text{on}} - \alpha_1 \nabla_{\phi_{\text{on}}} \hat{\mathcal{L}}(\tau_{\text{synth}})$ ;     `// Train the online model on Equation 10.`

$\quad \phi_{\text{off}} = \phi_{\text{off}} - \alpha_2 \nabla_{\phi_{\text{off}}} \sum_j \hat{\mathcal{L}}_{\text{off}}(\tau_e^j)$ ;     `// Train the offline model on Equation 11.`

$\quad \psi = \psi - \alpha_3 \nabla_\psi \sum_j \hat{\mathcal{L}}_{\text{off}}(\tau_e^j)$ ;     `// Train the dynamics model on Equation 11.`

$\quad \eta = \eta - \alpha_4 \nabla_\eta \sum_j \hat{\mathcal{L}}_{\text{off}}(\tau_e^j)$ ;     `// Train the policy on Equation 11.`

**end**

**Algorithm 3:** Training deconfounded imitators, offline variant

## D   Multi-armed bandit experiment

**Implementation details**   The inference model $q_\phi$ is implemented as an RNN with GRU architecture Cho et al. (2014) with a hidden layer of 256 units. Before the RNN, the observation is preprocessed by an

| Hyperparameter | Value |
|---|---|
| Episode length | 100 |
| Imitation training steps | 5000 |
| Dynamics model training batch size (full episodes) | 100 |
| Imitation training batch size (full episodes) | 100 |
| Behavioral cloning learning rate | 0.001 |
| Variational inference learning rate | 0.0001 |
| KL coefficient ($\beta$) | 0.001 |

Table 1: Hyperparameters for the deconfounded behavioral cloning and naive behavioral cloning algorithms

MLP with two hidden layers of size 256 units and output size 32. The action is preprocessed by a linear transformation to a 32 dimensional vector. The outputs of the RNN are processed by a linear transformation to a vector which parametrizes the latent distribution. The latent distribution is a 256 dimensional Gaussian. One half of the predicted vector represents the mean of the latent distribution and the other half, after softplus activation has been applied to it represents the variance.

The decoder is an MLP with two hidden layers of size 256 and a linear output layer. It uses the same input preprocessing networks for the observations and actions as the inference model. The policy is an MLP, which takes the latent sample, and an observation as inputs. It uses the same observation embedding network as the other networks and then has two hidden layers with 256 units each. The naive BC baseline uses the same network architecture as the deconfounded algorithm, except it does not represent the belief as a probabilistic latent variable and therefore there is no sampling step. It just directly passes output of the trajectory encoder as the input to the policy network. All of the MLPs use ReLU activations. All networks are optimized using the Adam optimizer (Kingma & Ba, 2015) with default settings from PyTorch (Paszke et al., 2019), except for the learning rate.

**Hyperparameter settings**  The hyperparameters used for the learning algorithms are presented in Table Table 1.

**Computing the ground truth policies**  All of the probabilities relevant to the bandit problem are known exactly from the definition of the problem and the conditional and interventional policies given in the main paper. Using these probabilities we can compute the true conditional and interventional policies, allowing us to compare the learned algorithms to the relevant optimal policies.

In practice, the true belief over theta can be computed for any trajectory as follows

$$\log p(\hat{\theta}_0) = \log \frac{1}{5}, \; \log p(\hat{\theta}_{t+1}[A_t]) = \begin{cases} \log p(\hat{\theta}_t[A_t]) + s_t \log \frac{3}{4} + (1 - s_t) \log \frac{1}{4} & \text{if } A_t = a_t \\ \log p(\hat{\theta}_t[A_t]) + s_t \log \frac{1}{4} + (1 - s_t) \log \frac{3}{4} & \text{otherwise} \end{cases}. \tag{12}$$

The true interventional policy can then be computed by sampling a belief $\hat{\theta}_t \sim \log p(\hat{\theta}_t)$, and sampling an action from $\pi_{\exp}(a_t|s_t, \hat{\theta}_t)$. This can be seen as a Thompson sampling policy (Thompson, 1933), which acts optimally given its current belief of the task. The true conditional policy is computed similarly, except taking the actions as evidence for the latent is added to the update

$$\log p(\hat{\theta}_{t+1}[A_t]) = \begin{cases} \log p(\hat{\theta}_t)[A_t] + s_t \log \frac{3}{4} + (1 - s_t) \log \frac{1}{4} + \log \frac{6}{10} & \text{if } A_t = a_t \\ \log p(\hat{\theta}_t)[A_t] + s_t \log \frac{1}{4} + (1 - s_t) \log \frac{3}{4} + \log \frac{1}{10} & \text{otherwise} \end{cases}. \tag{13}$$

## E   Confounded MDPs

**Environment descriptions**  For `LunarLander-v2` (Brockman et al., 2016), we consider a modified version with unknown key bindings: a latent $\theta$ specifies a permutation in the map between two of the the agent's actions and the behaviors (firing the left and right engines of the space craft). This permutation is known

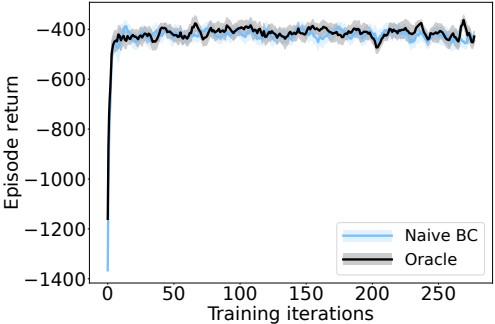

Figure 7: Comparing naive BC and oracle in a deterministic variant of `HalfCheetahBulletEnv-v0`. Most of the difference in the episodic returns here compared to the returns reported in Swamy et al. (2022b) follows from their results adding a constant 1000 to the episodic return for plotting.

to the expert, but not the imitator. In addition, we make the environment stochastic: at each step, with a probability of 0.15, a uniformly chosen random action is performed instead of the action chosen by the agent. The confounding problem arises because the expert's actions are temporally correlated, for example it may select the same action multiple times to adjust the lander's attitude. A naive recurrent BC policy may thus use past actions to predict future actions - causing confounding if the controls are swapped, as a wrong initial guess influences future decisions.

For `HalfCheetahBulletEnv-v0` (Coumans & Bai, 2016–2021), we modify the environment similarly to Swamy et al. (2022b) by varying the target speed. The expert observes the true target speed while the imitator only observes an indicator, which shows whether it is running faster or slower than the target speed. The expert smoothly tends to the target speed. This leads to correlated actions, in which the trend of past speeds can be extrapolated. A confounded naive BC imitator may attempt to extrapolate a random trend of its past speeds. Contrary to Swamy et al. (2022b), we find that the deterministic version of this environment does not necessarily differentiate between naive BC and DAgger. Therefore, we make the indicator stochastic by randomizing it 20% of the time. For a detailed discussion about the effect of stochasticity on `HalfCheetahBulletEnv-v0`, see below.

In `AntGoal-v0` (Todorov et al., 2012), we consider a version of the classic ant robot locomotion environment, where the task is to run to a goal randomly sampled from within a circle around the starting point. This is similar to the ant environment considered by Zintgraf et al. (2020). We make the goal radius $\frac{10}{3}$ times larger to make the locomotion task more challenging and remove the control cost. The imitators receive noisy observations of the length and angle of the vector between the ant and the goal in the global coordinate frame. Gaussian noise with scale 0.1 is added to the indicator. The expert knows the true goal location and starts moving toward it immediately, making its actions correlated across time. The confounding arises because on the expert trajectories, the goal direction and distance can be inferred from the expert actions and states without considering the noisy indicator.

**Deterministic vs stochastic variant of HalfCheetah.** We test the different imitation learning algorithms on the `HalfCheetahBulletEnv-v0` environment from Swamy et al. (2022b). We found that running our implementation of naive BC matches the performance of our implementation of DAgger, which is inconsistent with the results of Swamy et al. (2022b), where they report that DAgger outperforms naive BC. See figure Figure 7 for learning curves. One reason for why their results show a difference between the algorithms may be that their implementation of BC may suffer from covariate shift resulting from it having been trained on a small number of samples compared to DAgger. However, if that is the case, it is not an example of the confounding problem we are interested in. To make sure that we are testing for the ability of the algorithms to deal with the confounding problem in this environment, we make the indicator variable stochastic. The results for the stochatic variant are reported in the main paper.

| Hyperparameter | Value |
|---|---|
| Top-level training iterations | 300 |
| Imitation / inference training steps per top-level iteration | 100 |
| Dynamics model training batch size (full episodes) | 16 |
| Imitation training batch size (full episodes) | 16 |
| BC learning rate | 0.0003 |
| Variational inference learning rate | 0.0003 |
| KL coefficient ($\beta$) | 1.0 |

Table 2: Hyperparameters for the deconfounded BC, DAgger, and naive BC algorithms for `LunarLander-v2`, `HalfCheetahBulletEnv-v0`, and `AntGoal-v0` environments. We fix the episode length for `LunarLander-v2` to 500 steps, for `HalfCheetahBulletEnv-v0` to 1000 steps, and for `AntGoal-v0` to 200 steps.

**Implementation details**  For LunarLander, HalfCheetah, and Ant we use the same network architecture as the bandit experiments with the following changes. The output of the encoder parametrizes the mean and the log-variance of the latent distribution. There is no parameter sharing between the different input processing networks in the encoder, policy, and decoder. The RNN has a hidden size of 500. The MLPs have two hidden layers of size 128. The action embedding has size 128. The latent distribution is a 64 dimensional Gaussian for LunarLander and HalfCheetah, and 8 dimensional Gaussian for Ant. In practice, the latent dimensionality does not matter very much. The MLPs use ELU (Clevert et al., 2015) activations followed by LayerNorm (Ba et al., 2016). The networks are optimized with AdamW (Loshchilov & Hutter, 2017).

The experts are trained using a PPO implementation by Raffin et al. (2021) with hyperparameters from Raffin (2020). Additionally, a running normalization layer (Raffin et al., 2021) is applied to the observations and fixed after expert training to make the distribution of the observations easier to learn the decoder on.

On each iteration of the training algorithm, a batch of new episodes are collected with the appropriate policy or policies. Naive BC only collects data using the expert policy. DAgger collects data using a policy defined by a mixture of the expert and imitator policies, which is linearly annealed from the expert policy to the imitator policy during the training. Deconfounded algorithm collects a batch episodes with the expert policy for BC and a batch episodes with the imitation policy for training the encoder. These trajectories are saved in circular buffers that hold $2 * 10^6$ transitions. After each data collection step, the imitator (and encoder) are updated for 100 steps with the update function corresponding to the algorithm.

We implemented GAIL closely following a popular publicly available implementation and using recurrent PPO from `stable-baseline3` Raffin et al. (2021) as the RL algorithm. We use the same data sampling pipeline as the deconfounded imitation learning for GAIL. Deconfounded algorithm collects a batch episodes with the expert policy for BC and a batch episodes with the imitation policy for training the encoder and stores those samples in separate buffers. The network architectures are the same as for the other policies, where applicable. We found that getting GAIL to work at all in our environments, we had to increase the number of episodes sampled between each update from 16 to 64. Furthermore, we found that GAIL did not work with reward function parameterized as a function of $(s, a, s')$ in our environments, but worked better as a function of just $(s')$.

**Hyperparameter settings**  The hyperparameters used for the learning algorithms are presented in table Table 2.

| Hyperparameter | Value |
|---|---|
| Top-level training iterations | 300 |
| PPO training iterations per algorithm iteration | 1 |
| PPO batch size (timesteps) | 400 |
| PPO gradient max norm | 0.5 |
| PPO normalize advantage | True |
| PPO epochs per training iterations | 10 |
| PPO discount rate $\gamma$ | 0.99 |
| PPO GAE-$\lambda$ | 0.95 |
| PPO learning rate | 0.0003 |
| PPO episodes per training iteration | 64 |
| Discriminator training steps per top-level iteration | 20 |
| Discriminator batch size (full episodes) | 16 |
| Discriminator maximum gradient norm | 100 |

Table 3: Hyperparameter settings for GAIL.

