# OpenReview forum: "Deconfounding Imitation Learning with Variational Inference"
_TMLR — Accepted by TMLR_

### Review · Reviewer_robv · 2024-05-14

**Summary Of Contributions:**

This paper tackles the issues of confounding latent variables in imitation learning. In particular, this is where there are unknown facts that the expert is aware of that affect their policy, but the imitator is not. The authors propose a solution by learning a _interventional policy_ using variational inference. They make a theoretical case for this idea (under some assumptions) and showcase it in practice (where the assumptions are relaxed). They run experiments on a multi-armed bandit and various other more complicated environments.

**Audience:**

Yes

**Broader Impact Concerns:**

N/A.

**Claims And Evidence:**

No

**Requested Changes:**

- I think this paper requires a pass for grammar and typos. Further, the writing could be made clearer, for example, by breaking some long sentences up. For example:
    - Page 1: _This enables the agent to make an inference about the value of the **latent** based on everything it has observed so far, which eventually allows it to break ties over the actions in states where the **latent** matters_; latent what? Do you mean **latent variable**?
    - Page 4: _The goal of imitation learning in this setting is to learn imitator parameters η such that when the imitator is executed in the environment, the imitator agrees with the expert’s decisions, meaning we wish to maximise Eθ∼P(θ) Eτ∼Pη(τ;θ)[Pst,at∈τ logπexp(at | st,θ)]._ I think this sentence should be broken up.
    - Page 4: _We say **that current** state $s_t$_:  that **the** current...
    - Page 4: _next expert’s action at are confounded**,** because the distribution_: remove the comma.
    - In some equations with multiple terms (e.g. Equation 2 and Equation 3, can the spacing after commas be increased for readability?
    - Page 10: _The expert policy is defined in Eq. **equation** 2._: Remove **equation**.
    - Page 10: _The hyperparameters for the algorithms are provided in **App**. D._: In a previous part, you used **Appendix**. Be consistent! Similarly, for Figure(s) vs Fig(s).

- The following statements need to be supported better, either by another reference or a broader discussion:
     - Page 2: _This problem can be a source of “hallucinations” in generative models including large language models._
     - Page 2: _Acting upon a mistaken belief about the latent variable propels the agent out of the support of the expert data._
     - Page 5: _In a causal framework, this corresponds to treating the choice of past actions as interventions._
     - Page 10: _Even the naive learner needs a memory to be able to adapt to a new instance of the bandit defined by the latent._

### Suggested changes for clarity

- Page 2: _query access to the expert policy_: what does **query access to the expert policy** mean exactly? Since you are saying that your solution doesn't require this **query access** I think it would be good to describe this more. Query according to what?
- Page 4: _distribution over at when we intervene on $s_t$_: mathematically, what does it mean to **intervene$ on $s_t$?
- In Section 3.3, Equation 1, is $p(\theta)$ assumed to be known? Is this the same as $P(\theta)$ or is this a prior over the latent variables? If it is the latter, I think this should be stated in the text.
- Page 5: _A solution to this issue is to only take as evidence the data that was actually informed by the latent, which are the transitions._; This sentence is unclear. Could you expand on it?
- Page 6: _First, we assume that the latent variable θ of the MDP is **static**_: what exactly does **static** mean here? Do you mean that the latent variable is fixed within an episode? I think writing it out more explicitly would be better.
- Page 6: What is the definition of **recurrent** here? Does Assumption 1 state a definition for a **recurrent MDP** or is it stating an implication: if all pairs $s, a$ are reached infinitely often in each trajectory, then the MDP is recurrent; is this implication the definition of a recurrent MDP? I think I am confused by this statement being an **Assumption**.
- Section 4: What are the key practical takeaways from the theory?
- Page 10: _As we are only interested in the effects of confounding on imitation learning, and not in **finite-data effects**,_: what are some examples of **finite-data** effects?
- Figure 4: What do the sub-captions mean? It is confusing.

- Section 6: I think the experiments need to be described better. What exactly are the differences between the setups in Figure 4(a) and Figure 4(b)?

- In Figure 5, the legend should be moved outside of the figure bounding boxes (for example to be above all of them) since it is applied to all the sub-figures.

- It would be good to consistently use either **Naive BC** or **Naive Imitation Learning**.



### Minor changes
- In Figure 1, can you change the last sentence to: _The expert action depends on the latent variable θ **(red arrows)** whereas the imitator action does not_.
- Section 3.3 Heading: I think it would better to expand out BC to Behavioural Cloning.

**Strengths And Weaknesses:**

### Strengths
- In example of the self-driving car in the Introduction was insightful and provides a good grounding to understand the core problem that the paper tries to tackle.
- Sufficient summary of the literature.
- The theoretical justification (if correct -> I was not able to check the correctness of the theory) followed by experimental work is a nice progression.
- Running experiments on 10 random seeds is good.
### Weaknesses
See requested changes.

---

> ### Author Response · Authors · 2024-06-26
>
> Thank you for your thoughtful review. We respond to the requested changes below and implemented the changes in the updated version.
>
> ## Grammar and typo fixes
>
> Thank you for pointing out these issues. We implemented all of the suggested changes in the revised version of the paper.
>
> ## Changes for clarity
>
> > Page 2: query access to the expert policy: what does query access to the expert policy mean exactly? Since you are saying that your solution doesn't require this query access I think it would be good to describe this more. Query according to what?
>
> By query access we mean that the expert policy is available for choosing actions in states encountered during the imitation learning phase. This is in contrast to standard imitation learning where the expert policy is only available via a dataset of fixed samples. In the revised paper, we removed the query access term from the abstract and introduce it with more descriptive language in the introduction on the revised page 2.
>
> > Page 4: distribution over at when we intervene on $s_t$: mathematically, what does it mean to **intervene** on $s_t$?
>
> Confounding can be defined in two ways: by saying that there is a common ancestor in the probabilistic model, or by referring to a (hypothetical) intervention, as is done in (Peters et al, 2017, def 6.39). In the original manuscript, we used the latter definition, but we agree with the reviewer that this then requires more clarity what such an intervention means. To simplify the presentation, we use the former definition in the revised version.
>
> > In Section 3.3, Equation 1, is $p(\theta)$ assumed to be known? Is this the same as $P(\theta)$ or is this a prior over the latent variables? If it is the latter, I think this should be stated in the text.
>
> In the theory section 4.1 and in our practical algorithm, we do not assume that we know the prior $p(\theta)$. We denote the unknown latent variable of the environment as $\theta$ and the known inferred latent as $\hat \theta$. In equations (1) and (3) of the ground-truth conditional and interventional policies, we infer the theta using the ground-truth conditional probabilities, so we choose not to add a hat.
>
> > Page 5: A solution to this issue is to only take as evidence the data that was actually informed by the latent, which are the transitions.; This sentence is unclear. Could you expand on it?
>
> We replaced this sentence by “actually informed by the latent, which is the dynamics distribution $p(s_{t+1}  \mid s_t, a_t, \theta)$”.
>
> > Page 6: First, we assume that the latent variable θ of the MDP is static: what exactly does static mean here? Do you mean that the latent variable is fixed within an episode? I think writing it out more explicitly would be better.
>
> By the assumption of a static latent variable, we require the latent variable to stay unchanged for the duration of a single episode. This is necessary for the identification to work in theory. Note, that there is a tight coupling between the trajectories and MDPs in our setting since in practice, the agent interacts with each MDP for only one episode. We add a clarifying comment about this on page 6 where the term is introduced.
>
> > Does Assumption 1 state a definition for a recurrent MDP or is it stating an implication?
>
> We rephrased Assumption 1 in the revised version to state what we intended: we assume the MDP is recurrent.
>
> > Section 4: What are the key practical takeaways from the theory?
>
> That even though we don’t observe the latent, nor can query the expert, we can still infer the interventional policy. This is stated in the first two paragraphs of section 4.1. In section 3.6 we argue why it’s desired to infer this policy.
>
> > Page 10: As we are only interested in the effects of confounding on imitation learning, and not in finite-data effects,: what are some examples of finite-data effects?
>
> We removed the term “finite-data effects” and instead refer to over-fitting on a small dataset. In practice, behavioral cloning on a small expert dataset might not generalize to states the expert could have plausibly visited but didn't in the training dataset even when there's no confounding.
>
> > What exactly are the differences between the setups in Figure 4(a) and Figure 4(b)?
>
> We edited the sub-captions and the caption in the revision. The panel a) shows the probability the different policies assign to choosing the best arm when evaluated on data collected by the expert policy. The panel b) shows the same but on data collected by the policies themselves. And panel c) are learning curves where the y-axis is the reward.
>
> > In Figure 5, the legend should be moved outside of the figure bounding boxes (for example to be above all of them) since it is applied to all the sub-figures.
>
> We created a shared legend for Figure 5 and Figure 4.

---

> > ### Author Response · Authors · 2024-06-26
> >
> > > It would be good to consistently use either Naive BC or Naive Imitation Learning.
> >
> > For consistency, we changed all references to the naive behavioral cloning algorithm to consistently say “naive BC”, leaving “naive imitation learning” for the cases where a broader class of algorithms may be considered.
> >
> > > In Figure 1, can you change the last sentence to: The expert action depends on the latent variable $\theta$ (red arrows) whereas the imitator action does not.
> >
> > Updated the caption in the revision.
> >
> > > Section 3.3 Heading: I think it would better to expand out BC to Behavioural Cloning.
> >
> > Updated in the revision.
> >
> > ## Improving references and supporting discussion
> >
> > > This problem can be a source of “hallucinations” in generative models including large language models.
> >
> > We change the term in the introduction to “delusions” as that is the term Ortega et al. 2021 use and add the citation. For a broader discussion about causal confounding in behavioral cloning, see e.g., [8:50 to 15:40 on John Schulman’s lecture on RLHF](https://www.youtube.com/watch?v=hhiLw5Q_UFg).
> >
> > > Acting upon a mistaken belief about the latent variable propels the agent out of the support of the expert data.
> >
> > We removed the claim about the mistaken beliefs propelling the agent out of expert support from the revised version, as it is not required for introducing the benefits of DAgger in the introduction.
> >
> > > In a causal framework, this corresponds to treating the choice of past actions as interventions.
> >
> > We substantiate that claim in the sentences thereafter. We will add a linking clause in the revision to clarify. Or is the reviewer referring to the do-calculus reasoning that follows? This is quite a standard step, and already has a reference, but we’re happy to elaborate if the reviewer prefers.
> >
> > > Even the naive learner needs a memory to be able to adapt to a new instance of the bandit defined by the latent.
> >
> > For the naive baseline, we attempt to learn the conditional policy in equation (1). This is not a Markovian policy, because the history is used to infer the latent variable. An even more naive baseline could be to try to learn a Markov policy. At best, such a policy chooses the correct action in the bandit experiment with a probability of 1/#arms=0.2, which is worse than the non-Markov conditional policy (see fig 4b). We clarified this in the revised version.

---

### Review · Reviewer_QK3p · 2024-06-11

**Summary Of Contributions:**

This paper proposes to learn an imitation learning policy with partial observability of the environment. The authors first explain the naive BC algorithm, and how it would fail under partial observability. Then, it gives a theoretical analysis of how to learn an interventional policy and a practical implementation of it. The key is to approximate the state dynamics using an expert demo and learn an inference model that maps exploration trajectories to the confounding variable theta. In the experiment section, the authors show two sets of simulated experiments that show that the proposed algorithm recovers the interventional policy.

**Audience:**

Yes

**Claims And Evidence:**

Yes

**Requested Changes:**

1. Figure 3 is confusing. What's the difference between the dotted line and the solid line? Why is the exploration policy coming from the blue blocks on the right? Could the authors explain each component more clearly?
2. A more detailed discussion of the abovementioned weaknesses could strengthen the paper.

**Strengths And Weaknesses:**

Strengths:
1. Although the derivation of of the algorithm is based on strong assumptions, the authors validate the main assumptions in the experiment section.
2. The motivation of this method is well-explained, particularly with the example in section 3.3
3. The proposed method is intuitive and supported by experiments.

Weaknesses:
1. Some details of the method are not clear:
    1) As described in 4.1, the identifiability of the interventional policy is based on several crucial assumptions. One assumption is that the latent variable theta is static. Can the authors elaborate on what that means? Static with respect to the environment or static with respect to a single trajectory?
    2) How is the exploration policy obtained?
    3) How many expert demonstrations are required to learn the approximated dynamics model for each environment?
    4) In the experiment section, how are the true interventional policy and the true conditional policy obtained?
2. In Figure 5, the proposed method is generally worse than DAgger. Can the authors include an analysis of the benefits of this method over DAgger?
3. The authors discussed this method's limitations in section 7. A more detailed discussion about how this method would involve solving real-world tasks with confounding variables is encouraged.

---

> ### Author Response · Authors · 2024-06-26
>
> Thank you for your thoughtful review. We respond to the requested changes below and implemented the changes in the updated version.
>
> ## Clearer figure 3
> The blue block on the right represents a policy that is conditioned on the inferred latent. In practice, we found it convenient to use this policy as the exploration policy for generating data for training the inference model.
>
> The dotted line for the policy should be interpreted in contrast to the solid line for the inference module. The solid line represents that specifically the inference model $ q(\hat{\theta} \mid \tau) $ has to be used in the policy training. In contrast, the policy $\pi(a \| s, \hat{\theta})$ is used for the exploration policy only as a convenience but any other recurrent policy could be used instead.
>
> Thanks for pointing these issues out. We will edit the caption to explain the figure more clearly.
>
> ## Comparison to DAgger
>
> DAgger indeed performs better than our method, because it can be seen as an oracle, which has access to more information than the other methods, namely the expert’s action on states encountered by the imitator. As discussed in [Ortega, 2021], the DAgger policy converges to the interventional policy. We will clarify this in the revised paper.
>
> ## Clarifications
> > Can the authors elaborate on what [static latent variable] means?
>
> By the assumption of a static latent variable, we require the latent variable to stay unchanged for the duration of a single episode. This is necessary for the identification to work in theory. Note that alternatively this can be interpreted as having a different fully observed MDPs, one for each value of the latent variable, and the imitator not knowing which MDP it interacts with. We add a clarifying comment about this on page 6 where the term is introduced.
>
> > How is the exploration policy obtained?
>
> In theory, the exploration policy can be any recurrent policy. In practice, as stated in section 5, we use the imitator's policy conditioned on the latent inferred by the inference model. We re-iterate this comment in the updated caption of Figure 3.
>
>  > How many expert demonstrations are required to learn the approximated dynamics model for each environment?
>
> The dynamics model training does not require expert demonstrations. Instead it trains on data from the exploration policy. In our practical experiments, a hundred new episodes are drawn in each iteration of the algorithm. As we are focused on the confounding problem, we did not attempt to make the model more data efficient but ran it with parameters that worked smoothly.
>
> > In the experiment section, how are the true interventional policy and the true conditional policy obtained?
>
> For the bandit experiments, because of the small finite state and action spaces, we can explicitly compute the true conditional and interventional policies via equations (1) and (3). Details are given in appendix D. This computation can be seen as an instance of theforwards algorithm (see e.g., Murphy, Kevin P. Machine learning: a probabilistic perspective. MIT press, 2012. page 609).
>
> ## Discussion on solving real world tasks
>
> In principle, the method consists of well-behaved components (amortized variational inference like in a VAE and behavioral cloning), which are known to scale. However, we did not focus on the data efficiency of the method, so directly using it as described in the paper would likely not work very well for real world problems. We expanded the discussion on limitations in the revised version of the paper.

---

### Review · Reviewer_ZAFY · 2024-06-19

**Summary Of Contributions:**

This paper considers imitation learning problems when the expert demonstrators have different sensory inputs than the imitating agent. In other words, the expert knows more about the world than the imitator. The main challenge comes from the hidden confounders in the casual graph caused by partial observability. To address this challenge, this paper proposes to train a variational inference model to train a latent-conditional policy. The proposed method is evaluated in the multi-armed bandit problem and multiple control environments.

**Audience:**

Yes

**Claims And Evidence:**

Yes

**Requested Changes:**

1.	This work makes a strong assumption that the latent variable is constant in each trajectory. This assumption makes the dynamics and expert policy stationary in each trajectory and the interventional policy becomes identifiable. However, this strong assumption may limit the generalization of the proposed algorithm.

2.	There is no clear definition of Confounded repeated multi-armed bandit and Confounded sequential decision making problems.

3.	The experiments were only conducted in low-dimensional environments.

**Strengths And Weaknesses:**

1.	This paper studies the challenging problem where naïve imitation learning algorithms fail in the presence of latent confounders. To address the challenge, this work first studied under which conditions latent confounding impacts the performance of BC. Informed by theoretical results, this work proposed a practical algorithm for deconfounding imitation learning with amortized variational inference. The experiments in multi-armed bandits and multiple control environments demonstrated that the proposed method can learn interventional policies.

2.	The proposed algorithm can get around the confounding problem without query access to the expert or IRL. In addition, the algorithm converges to the expert’s policy in theory, which means the expert’s policy can be identified under strong assumptions. The latent variable was inferred using variational inference and used for training imitation learning policies.

---

> ### Author Response · Authors · 2024-06-26
>
> Thank you for your thoughtful review. We respond to the requested changes below and implemented the changes in the updated version.
>
> > This work makes a strong assumption that the latent variable is constant in each trajectory. … However, this strong assumption may limit the generalization of the proposed algorithm.
>
> We appreciate that assuming a constant latent variable is a somewhat strong assumption. However, it is not clear what change the reviewer is asking us to make. First, in the submitted version of the paper, this is already listed as a limitation of the proposed method. Second, assuming a constant latent variable is common in other settings where a controller is learned. For example see [contextual MDPs](https://arxiv.org/abs/1502.02259), which are extensively used in the study of [generalization in RL](https://arxiv.org/abs/2111.09794).
>
> > There is no clear definition of Confounded repeated multi-armed bandit and Confounded sequential decision making problems.
>
> In the updated version of the paper, we changed the subsection titles in the empirical section to be less ambiguous and more descriptive of the content of the experiment.
>
> > The experiments were only conducted in low-dimensional environments.
>
> We agree with the reviewer that this is a sensible next step for our line of work. This limitation of the current work is already acknowledged in the paper. Furthermore, we conducted experiments in environments of similar complexity as other papers tackling confounding issues (see e.g., Swamy 2022b). Nevertheless, we expanded the discussion on the limitations of our evaluation and how the method is not directly applicable to real world problems.

---

### Decision · Action_Editor_4b6R · 2024-08-19

**Recommendation:** Accept as is

**Comment:**

The reviewers provided detailed suggestions for revisions, which have been addressed by the authors' responses. My recommendation is based on the fact that no further revisions were suggested in the final recommendations.

**Audience:**

The paper addresses an interesting problem for the imitation learning community.

**Claims And Evidence:**

The claims are accurate (under strong assumptions).